# An Assessment of the Nutritional Status in Patients with Inflammatory Bowel Disease—A Matched-Pair Case–Control Study

**DOI:** 10.3390/nu17081369

**Published:** 2025-04-17

**Authors:** Małgorzata Godala, Ewelina Gaszyńska, Konrad Walczak, Ewa Małecka-Wojciesko

**Affiliations:** 1Department of Nutrition and Epidemiology, Medical University of Lodz, 90-752 Lodz, Poland; ewelina.gaszynska@umed.lodz.pl; 2Department of Internal Medicine and Nephrodiabetology, Medical University of Lodz, 90-549 Lodz, Poland; konrad.walczak@umed.lodz.pl; 3Department of Digestive Tract Diseases, Medical University of Lodz, 90-419 Lodz, Poland; ewa.malecka-panas@umed.lodz.pl

**Keywords:** Crohn’s disease, ulcerative colitis, nutritional assessment, sarcopenia, body composition

## Abstract

Methods used in daily clinical practice for the assessment of the nutritional status in patients with inflammatory bowel disease (IBD) are often based on simple indices and may not be sufficient in the case of minor or early changes. The purpose of this study was to analyze the nutritional status in patients with IBD. Material and methods: The case–control study included 80 patients with IBD. The control group consisted of 80 healthy subjects matched based on age and gender. Body composition was measured using the electrical bioimpedance method. Results: Compared to the healthy women, the female patients with IBD had a significantly lower muscle mass (24.4 kg vs. 27.9 kg) and muscle strength (22.4 kg vs. 25.9 kg), as well as a lower MMI (7.8 kg/m^2^ vs. 9.9 kg/m^2^). Based on these findings, sarcopenia was diagnosed in 37.5% of the female patients, significantly more often than in the control group. In the group of men, there were no significant differences between the healthy controls and patients in terms of body composition and the prevalence of underweight and sarcopenia. Conclusions: The patients with IBD were characterized by a poorer nutritional status than the healthy subjects, mainly in terms of fat-free body mass and muscle mass, and consequently a higher incidence of sarcopenia, especially in the female group.

## 1. Introduction

Inflammatory bowel disease (IBD), such as Crohn’s disease (CD) and ulcerative colitis (UC), is one of the chronic autoimmune disorders whose incidence rates have been increasing for several decades, both in Poland and worldwide [1,2,3,4,5,6]. New findings are emerging in the pathogenesis of IBD concerning the melanocortin system. The melanocortin system, a complex set of molecular mediators of inflammatory processes, has gained importance in the pathogenesis and treatment of IBD. The results of studies conducted in mouse models suggest a potential inhibitory effect on colitis and participation in the complex cytokine imbalance of the intestinal microenvironment affected by chronic inflammatory damage [7]. The main symptoms include diarrhea, blood and mucus in stool, crampy abdominal pain, fever, and loss of appetite and body weight [7,8,9]. These symptoms can impact the nutritional status of patients with IBD and consequently affect both their well-being and the success of therapy.

Methods used in daily clinical practice for the assessment of the nutritional status in patients with IBD are often based on simple indices, such as body mass index (BMI) or unintentional weight loss. These methods, however, may not be sufficient in the case of minor or early changes in the patient’s nutritional status. Also, individuals with a high BMI, which is observed more and more frequently [10,11,12,13], may be classified as well-nourished or even over-nourished. Meanwhile, in patients with a high body weight relative to height, there often occurs a decline in muscle mass, which cannot be determined by estimating BMI alone. Therefore, body composition analysis seems to be a necessary part of the nutritional status assessment in patients with IBD.

More than a dozen diagnostic methods are available for a comprehensive and non-invasive assessment of the nutritional status. These include an analysis of body composition (BC), which plays an important role in measuring the content of individual body components, including fat mass, muscle mass, bone mass, and body water. Among the available methods, direct measurement methods of adipose tissue, such as computer tomography (CT) or magnetic resonance imaging (MRI), have been recognized as reference methods. However, these examinations are costly and have a certain impact on the patient’s health, and their results are largely dependent on the correct setting of the measurement view of the equipment. For this reason, other less invasive techniques, including dual-energy X-ray absorptiometry (DXA) or bioelectrical impedance analysis (BIA), have recently attracted increasing interest [8,14,15].

The DXA technique is less expensive than CT and MRI, and the X-ray dose to which the patient is exposed is about 20 times lower than in a routine chest X-ray. However, the technique requires specialized and rather costly measurement equipment as well as properly trained medical personnel [14,15]. The BIA technique is definitely less invasive. It involves the measurement of the impedance (composed of resistance and reactance) of body tissues, through which an electric current of low intensity but significant frequency is passed [16,17]. The BIA technique has been found to be a sufficiently accurate yet minimally invasive method that does not require the involvement of properly trained medical professionals, which has been confirmed in epidemiological studies [15,16,18].

The purpose of this study was to analyze the nutritional status in patients with inflammatory bowel disease based on body composition compared to a sex-matched and age-matched control group.

## 2. Materials and Methods

### 2.1. Study and Control Group

The case–control study included patients with IBD (40 women and 40 men) treated at the Department of Digestive Tract Diseases of the Medical University of Lodz. In the patients with CD, disease activity was assessed using the Crohn’s disease activity index (CDAI), whereas in the patients with UC, the Partial Mayo Score was applied [19,20]. The inclusion criteria for patients with IBD were age (18–80 years), outpatient treatment, and endoscopic and histological diagnosis of CD or UC. The exclusion criteria were the presence of prostheses/implants and pacemakers, occurrence of epilepsy, and abnormal conditions of the limbs or trunk (e.g., amputations, scoliosis, skin lesions).

The control group consisted of 80 healthy subjects matched based on age and gender. During the matching procedure, the weight and body mass index (BMI) of the respondents were not taken into account as factors that could affect body composition. This is an accepted practice confirmed in studies by other authors [21,22,23].

The study was conducted in accordance with the Declaration of Helsinki and approved by the Bioethics Committee of the Medical University of Lodz (No. RNN/70/22/KE, 14th June 2022). All the subjects gave written consent to participate in the study.

### 2.2. Anthropometry

All the subjects were measured for height (to the nearest 0.5 cm) and body weight (to the nearest 0.1 kg) using a stadiometer and a calibrated medical scale to determine body mass index (BMI). Underweight was diagnosed for a BMI < 18.5 kg/m^2^, normal weight—for a BMI from 18.5 to 24.9 kg/m^2^, overweight—from 25.0 to 29.9 kg/m^2^, and obesity—from 30.0 kg/m^2^ [24].

Body composition was measured using the electrical bioimpedance method with an InBody 270 device. In accordance with the generally accepted standards [stall], the subjects were fasted and asked to refrain from alcohol consumption and undertaking much physical activity for the 24 h preceding the study. Immediately prior to the measurement, the study participants were asked to empty their bladders, strip down to light underwear, and remove any jewelry to ensure a more precise measurement. Based on the obtained results, the following parameters were determined: total body water (TBW), fat mass (FM) and free fat mass (FFM), skeletal muscle mass (SMM), and protein and mineral content. Using the data obtained, the muscle mass index (MMI) was calculated as the ratio of total skeletal muscle mass to the square of height [kg/m^2^]. When assessing a significant reduction in muscle mass, men < 8.6 kg/m^2^ and women < 6.2 kg/m^2^ were used as cutoff points [25].

Handgrip strength was assessed using a SAEHAN digital hand dynamometer, No. DHD-1(SH1001). Measurements were taken twice, with the better result of the dominant hand being taken as the correct value [Alley]. Cut-off points for low muscle strength were adopted according to the criteria of the European Working Group on Sarcopenia in Older People 2 (EWGSOP2), for women < 16 kg and for men < 27 kg. Sarcopenia was diagnosed according to the EWGSOP2 criteria, with the presence of both low muscle strength (HGS) and low muscle mass (MMI) [25].

### 2.3. Physical Activity

The physical activity level of the respondents was assessed using the International Physical Activity Questionnaire (IPAQ). The questions included in the questionnaire focused on the frequency and timing of activity and were used to determine high (>1500 METs), moderate (600–1500 METs), or low (≤600 METs) level of physical activity among the respondents [26].

### 2.4. Dietary Intake

In all the subjects, the intake of energy and all nutrients was assessed based on a 24-h interview, taken three times from each study participant, by the dietetics staff. The average intake of energy and all nutrients was assessed using the computer program Diet 6.0 (license No. 52/PD/2022). The energy requirements of the patients were calculated using the Mifflin formula by multiplying the basal metabolic rate by the physical activity factor. An energy intake of less than 90% of the total energy requirement was considered inadequate [27].

### 2.5. Statistical Analysis

The data obtained were presented as mean values, standard deviations (SDs), and percentages. The distribution of the analyzed factors was verified using the Shapiro–Wilk test. Differences in BMI and body composition parameters between the groups were analyzed using the Mann–Whitney U test. For multiple comparisons, the Bonferroni correction was used. Relationships between the studied variables were evaluated using Spearman’s coefficient (r). All the multivariate models were controlled for the study participants’ age, level of education, family burden, and disease duration. To identify factors associated with sarcopenia, univariate and multivariate analyses were used. The assumed significance level was *p* < 0.05.

## 3. Results

The general characteristics of the subjects are shown in Table 1. The study included 40 women (50%) and 40 men (50%) with IBD. The mean duration of disease in the female and male groups was 8.9 and 8.7 years, respectively. Of all the patients, a majority received biological treatment (n = 64; 80%) and oral 5-aminosalicylic acid preparations (n = 64; 80%), while nearly one in three took immunosuppressive drugs (n = 30, 37.5%) and steroids (n = 25, 30.5%). Among the subjects with IBD, 34 patients were in clinical and endoscopic remission (42.5%). The groups of patients and healthy controls did not differ significantly in terms of age, sex, level of physical activity, or frequency of smoking.

Anthropometric analysis showed significant differences in the nutritional status between the female patients with IBD and healthy women. Compared to the healthy controls, the group of female patients presented a significantly lower mean BMI (22.6 kg/m^2^ vs. 25.5 kg/m^2^) and a higher prevalence of underweight assessed based on BMI (20% vs. 5%). Measurements of body composition in the subjects showed significantly lower levels of protein (6.7 kg vs. 8.1 kg), minerals (3.3 kg vs. 4.5 kg), and fat-free body mass (49.8 kg vs. 53.9 kg) in the individuals with IBD than in the control group. Compared to the healthy women, the female patients with IBD had a significantly lower muscle mass (24.4 kg vs. 27.9 kg) and muscle strength (22.4 kg vs. 25.9 kg), as well as a lower MMI (7.8 kg/m^2^ vs. 9.9 kg/m^2^). Based on these findings, sarcopenia was diagnosed in 37.5% of the female patients, significantly more often than in the control group.

In the group of men, there were no significant differences between the healthy controls and patients in terms of BMI value, body composition, prevalence of underweightness, and sarcopenia. The subjects differed only in terms of muscle strength, with lower values found in the group of patients than the control subjects (26.9 kg vs. 28.7 kg).

When evaluating energy and nutrient intake, significant differences were identified between the groups of patients and healthy subjects. A significantly lower energy intake was observed among the patients with IBD compared to the healthy subjects in the female group (1423.5 kcal vs. 1792.4 kcal). The female study participants also differed in their intake of proteins, fats, and carbohydrates, with a significantly lower dietary supply of these components in the group of patients. No significant differences in energy or nutrient intake were found in the male group (Table 1).

The study evaluated the effects of disease duration, its course, physical activity level, and energy and protein intake on body composition parameters. An inverse correlation was found between disease duration and fat-free body mass expressed in kilograms (r = −0.18, *p* < 0.05), muscle tissue content (r = −0.21, *p* < 0.05), MMI (r = −0.22, *p* < 0.05), and handgrip strength (r = −0.24, *p* < 0.05) (Table 2). Additionally, the subjects who had suffered from the disease for more than seven years showed a significantly lower mean fat-free body mass (53.2 kg vs. 58.9 kg), muscle mass (29.9 kg vs. 38.7 kg), MMI (7.9 kg/m^2^ vs. 10.1 kg/m^2^), and handgrip strength (21.6 kg vs. 27.1 kg) (Table 3).

There was a significant effect of physical activity undertaken on the body composition of the subjects. A positive correlation was found between the level of physical activity and fat-free body mass expressed both in kilograms (r = 0.21; *p* < 0.05) and as a percentage of body mass (r = 0.19; *p* < 0.05). Additionally, there was a positive correlation between physical activity level and muscle mass content (r = 0.23; *p* < 0.05), MMI (r = 0.22; *p* < 0.05), and handgrip strength (r = 0.32; *p* < 0.05). The study found significant differences in the nutritional status of patients with low and moderate/high levels of physical activity. Thus, patients with IBD undertaking low physical activity had a significantly higher mean BMI compared to those with moderate and high physical activity (24.7 kg/m^2^ vs. 22.1 kg/m^2^). Patients with IBD undertaking low physical activity were characterized by significantly higher body fat (21.1 kg vs. 18.3 kg) and significantly lower fat-free body mass expressed in kilograms (52.9 kg vs. 59.1 kg) and percentage of body weight (71.3% vs. 79.3%) compared to physically active patients. Patients with IBD undertaking low physical activity compared to those with at least moderate levels of physical activity were characterized by a lower skeletal muscle mass (30.9 kg vs. 36.4 kg), MMI (8.1 kg/m^2^ vs. 9.8 kg/m^2^), and handgrip strength (22.1 kg vs. 27.2 kg).

The study showed a significant impact of energy and protein intake on the nutritional status of the subjects. There was a positive correlation between energy intake and BMI (r = 0.25; *p* < 0.05) and handgrip strength (r = 0.24, *p* < 0.05). However, a comparison of patients with IBD and insufficient energy intake to those with normal energy intake showed a significantly lower mean BMI (21.2 kg/m^2^ vs. 23.9 kg/m^2^), lower body fat (18.3 kg vs. 21.9 kg), and lower handgrip strength (21.8 kg vs. 28.1 kg).

Protein intake positively correlated with fat-free body mass expressed in kilograms (r = 0.24, *p* < 0.05), muscle mass (r = 0.29, *p* < 0.05), MMI (r = 0.28, *p* < 0.05), and handgrip strength (r = 0.31; *p* < 0.05). Additionally, mean fat-free body mass (53.1 kg vs. 58.9 kg), muscle mass (30.7 kg vs. 38.2 kg), MMI (8.2 kg/m^2^ vs. 9.9 kg/m^2^), and handgrip strength (22.3 kg vs. 27.7 kg) were significantly lower compared to patients with normal protein intake.

The study also evaluated the effect of disease activity and disease type on the measured parameters of body composition. Compared to patients in remission, individuals with the active form of the disease had lower mean values for fat-free body mass (52.3 kg vs. 58.1 kg), muscle mass (31.5 kg vs. 37.9 kg), MMI (8.1 kg/m^2^ vs. 9.5 kg/m^2^), and handgrip strength (21.6 kg vs. 28.2 kg). The patients with CD compared to those with UC had a significantly lower BMI (20.9 kg/m^2^ vs. 23.7 kg/m^2^) and fat-free body mass expressed in both kilograms (52.2 kg vs. 58.2 kg) and percentage of body mass (73.7% vs. 76.9%). The subjects with CD, compared to those with UC, had a lower skeletal muscle mass (28.9 kg vs. 37.1 kg), MMI (7.8 kg/m^2^ vs. 9.9 kg/m^2^), and handgrip strength (21.9 kg vs. 26.9 kg).

Disease activity (OR = 0.636, 95% CI 0.462–0.839; *p* = 0.0189) and gender (OR = 0.663, 95% CI 0.471–0.874; *p* = 0.0191) were shown to be predictors of sarcopenia in the IBD patients after adjusting for age, BMI, and disease duration (Table 4).

## 4. Discussion

In our study, we assessed the nutritional status of patients with IBD using the electrical bioimpedance analysis (BIA). We showed significant differences in fat-free body mass and muscle mass in the group of women with IBD, which were significantly lower than in the control subjects. However, no significant differences in body composition were observed between the men with IBD compared to the control group. In the literature, there are few studies evaluating body composition in patients with IBD using electrical bioimpedance.

In a study by Geerling et al. [28], conducted in a group of patients with CD and UC, the authors noted a significantly lower fat-free body mass in female patients compared to healthy women, and no such differences were observed in the male group. These results are consistent with those of our study.

Similarly, in a study by Valentini et al. [29], male patients with IBD did not differ significantly from healthy subjects in terms of fat-free body mass. In contrast, significant differences were observed in the group of women, with a lower content of fat-free body mass recorded among subjects with IBD compared to healthy women. The results of this study confirm the findings of our research.

In a study by Głąbska et al. [21], conducted in a group of men with UC, the authors found no significant differences in body composition between patients and control subjects. The results of this study confirm the results of our findings, in which the nutritional status of male patients did not differ significantly from that of healthy subjects. However, Głąbska et al. [21] found a higher proportion of adipose tissue with a lower proportion of fat-free mass, which resulted in an increase in BMI above reference values in the studied patients with UC. We did not observe this relationship in our study. This discrepancy may result from the fact that, in the cited study, all subjects had UC and were in remission, which corresponds to a better nutritional status.

Different data were obtained in a study by Prieto et al. [30], in which patients with IBD had a higher fat-free body mass and muscle mass compared to the control group. These differences may be due to the specificity of the study group, in which the mean BMI of the patients was higher than in the control group and higher than in our study. Moreover, the mean BMI was indicative of overweightness, and more than 30% of the subjects were overweight or obese.

In the present study, the recruited control group was age-matched, not BMI-matched, and the BMI in the group of female patients was significantly lower than in the healthy women, whereas among the men, it was not significantly different. There are several possible reasons for the lack of differences in BMI. Firstly, in our study, the male group was dominated by patients with UC, for which aggravation of symptoms, such as diarrhea and weight loss, is less common, as confirmed in other studies [21,29,31,32]. The authors have often shown a higher BMI in patients with UC compared to control groups, and even a higher prevalence of overweight and obesity in these individuals [33,34,35,36,37]. Moreover, the men we studied had energy and nutrient intakes comparable to the control group, which indicates that these patients did not implement dietary restrictions and, consequently, had a normal BMI. Additionally, potential biological or hormonal factors (e.g., testosterone levels) might have some protective impact on muscle mass in males. In contrast, in the group of female patients, those with CD and active disease predominated, which may explain the lower mean BMI. Additionally, we found a significantly lower intake of energy and essential nutrients, including protein, which is crucial for building muscle mass and fat-free body mass.

We showed differences in the body composition between the groups of patients with CD and UC. The former had a significantly lower BMI, body fat, and fat-free body mass, as well as muscle mass. Handgrip strength was also lower in the group of patients with CD than in those with UC. The results of studies available in the literature partially confirm our findings. Various authors have shown a significantly lower BMI and body fat mass in patients with CD than in those with UC, which is in line with the results of our study. However, they indicated that there were no differences in fat-free body mass, which does not confirm the results of our study [38,39,40]. There are also papers that report no significant differences in body composition between patients with CD and UC [29]. The issue of the effect of disease type on the nutritional status of patients with IBD, particularly body composition, has not yet been resolved and requires further study. More research is needed to explain the discrepancies in the results obtained by the authors of studies demonstrating significant differences in the nutritional status between patients with CD and UC and others who did not observe any such differences.

In our study, we proved a significant effect of disease activity on the nutritional status and body composition in patients with IBD. We showed that individuals in remission were characterized by a higher fat-free body mass compared to those with active disease. Similar data were also presented by other authors.

A study by Kim et al. [41] found that disease activity significantly determined body composition of patients with IBD and showed that those in remission were characterized by a higher fat-free body mass and muscle mass compared to patients with exacerbation of symptoms.

Also, in a study by Prieto et al. [30], patients in remission had a significantly higher fat-free body mass compared to those with active disease.

The results of the cited studies indicating a better nutritional status among patients with IBD in remission seem consistent and logical. The better nutritional status in this phase of the disease probably results from a higher dietary intake of energy and nutrients and lower losses that accompany diarrhea present in the active phase of the disease.

We also showed an association of disease duration with the nutritional status of the subjects. We found a negative correlation with fat-free body mass and muscle mass. The data available in the literature on the relationship between disease duration and body composition of patients with IBD are inconsistent. Some authors have not confirmed any relationship between disease duration and body fat and fat-free body mass [29,32,42]. Others, however, have shown such a relationship.

For example, a study by Jahnsen et al. [40] showed that patients with UC (not those with CD, though) who had suffered from the disease for more than seven years had a lower fat-free and fat body mass compared to patients with a shorter duration of the condition. This relationship was found despite the longer total duration of steroid treatment in patients with CD.

A study by Geerling et al. [32] compared the body composition of patients with newly diagnosed IBD and patients with CD with a long history of the disease. The authors reported a significantly lower fat-free body mass, body fat mass, and BMI in patients on long-term treatment.

In a study by Schneider et al. [43] conducted among patients with CD in remission, a negative correlation was observed between disease duration and fat-free body mass. These findings confirm the results of our study.

The issue that has been increasingly discussed in the literature is the higher risk of sarcopenia among patients with IBD due to chronic inflammation, dietary restrictions, and long-term pharmacotherapy. In our study, sarcopenia was diagnosed in one in five patients, significantly more frequently in the female group. The results we obtained confirm the available data according to which the incidence of sarcopenia in patients with IBD ranges from several to even several tens of percent, depending on the definition adopted and methods of body composition assessment used [44,45]. According to the latest EWGSOP2 recommendations, sarcopenia is diagnosed along with the identification of low muscle strength and muscle mass [25]. Studies have confirmed that handgrip strength is a better predictor of low fat-free body mass and muscle mass than BMI, which is more correlated with body fat and may indicate a false negative result in the diagnosis of sarcopenia [46,47]. Therefore, handgrip strength test along with BMI measurement should be routinely performed in patients at increased risk of sarcopenia.

The present study found a significa”tly ’ower handgrip strength in the group of patients with IBD compared to healthy subjects. This relationship also applied to the group of men, in whom body composition analysis showed no significant differences in fat-free and muscle mass. This confirms the EWGSOP recommendations regarding the need to measure handgrip strength in patients with IBD independently of body composition assessment. Data available in the literature are congruent with the results of our study. The authors indicated a significantly lower muscle strength among patients with IBD compared to healthy subjects [9,46,47,48,49].

An interesting issue is the effect of regularly undertaken physical activity in the prevention of sarcopenia. In our study, we showed an association between the level of physical activity and fat-free body mass, muscle mass, and handgrip strength. The patients undertaking at least moderate physical activity showed a higher content of fat-free body mass, muscle mass, as well as greater muscle strength. Additionally, patients with IBD often engage in reduced regular physical activity due to fear of disease exacerbation and, frequently, a lack of encouragement from their social network [50]. There are studies available that, on the one hand, indicate limitations of patients with IBD in undertaking regular physical activity and, on the other hand, report improved quality of life in physically active individuals affected by this condition [50,51,52].

Our study has some limitations. It is a single-center study, and, therefore, may not be representative of all patients with IBD. It was conducted in a relatively small group of patients, in which most of the participants were receiving biological therapy, and the control group consisted of healthy subjects only.

## 5. Conclusions

In our study, the patients with IBD were characterized by a poorer nutritional status than the healthy subjects, mainly in terms of fat-free body mass and muscle mass, and, consequently, a higher incidence of sarcopenia, especially in the female group. In the diagnosis of sarcopenia, clinical assessment based on BMI alone is insufficient; therefore, muscle strength testing should be included in the routine anthropometric evaluation of patients with IBD because of its association with low muscle mass. A comprehensive, multicenter study on the nutritional status in individuals with IBD is required to ensure the early diagnosis of malnutrition and sarcopenia and improve the quality of life and treatment outcomes among these patients.

## Figures and Tables

**Table 1 nutrients-17-01369-t001:** General characteristics of the study participants.

Characteristics	Women		Men	
	IBD Mean ± SD/n(%)	Healthy Mean ± SD/n(%)	IBD Mean ± SD/n(%)	Healthy Mean ± SD/n(%)
Age [years]	37.5 ± 3.6	37.9 ± 3.3	40.3 ± 3.1	40.1 ± 2.9
Body height [cm]	167.7 ± 4.7	165.3 ± 4.1	179.5 ± 6.9	177.8 ± 7.3
Body mass [kg]	65.4 ± 6.1	69.8 ± 5.9	78.8 ± 10.2	79.7 ± 11.3
Smokers	5 (12.5)	4 (10)	9 (22.5)	8 (20)
CD	30 (75)	-	16 (40)	-
Duration of disease [years]	8.9 ± 5.3	-	8.7 ± 4.1	-
Treatment				
Biological treatment	34 (85)	-	30 (75)	-
Immunosuppressive therapy	15 (37.5)	-	15 (37.5)	-
Steroids	15 (37.5)	-	10 (25)	-
5-ASA	34 (85)	-	30 (75)	-
Disease activity				
Active	26 (65)	-	20 (50)	-
Remission	14 (35)	-	20 (50)	-
Anthropometry				
BMI [kg/m^2^]	22.6 ± 4.2 *	25.5 ± 3.6 *	25.1 ± 3.4	26.3 ± 3.1
<18.5	8 (20) *	2 (5) *	-	-
18.5–24.9	18 (45)	18 (45)	20 (50)	17 (42.5)
25.0–29.9	11 (27.5)	12 (30)	17 (42.5)	20 (50)
≥30	3 (7.5) *	10 (25) *	3 (7.5)	3 (7.5)
Proteins [kg]	6.7 ± 1.2 *	8.1 ± 1.4 *	9.9 ± 2.1	10.3 ± 2.8
Minerals [kg]	3.3 ± 0.7 *	4.5 ± 0.9 *	6.9 ± 3.1	7.7 ± 3.3
TBW [kg]	35.1 ± 4.9	34.4 ± 5.2	46.4 ± 4.4	44.7 ± 5.1
FM [kg]	20.3 ± 4.4	20.5 ± 5.3	18.4 ± 6.6	19.2 ± 5.9
FM [%]	22.1 ± 3.5	23.3 ± 4.1	22.4 ± 5.6	24.3 ± 4.9
FFM [kg]	49.8 ± 5.1 *	53.9 ± 6.7 *	61.5 ± 6.2	59.6 ± 7.1
FFM [%]	76.4 ± 6.8	77.7 ± 4.8	77.1 ± 5.1	75.7 ± 6.6
SMM [kg]	24.4 ± 2.7 *	27.9 ± 3.1 *	39.3 ± 4.1	39.8 ± 3.7
Handgrip strength [kg]	22.4 ± 3.2 *	25.9 ± 3.3 *	26.9 ± 2.7 *	28.7 ± 4.1 *
Low	20 (50)	2 (5)	10 (25)	2 (5)
MMI [kg/m^2^]	7.8 ± 3.3 *	9.9 ± 2.5 *	9.1 ± 2.7	11.5 ± 2.4
Sarcopenia	15 (37.5) *	2 (5) *	3 (7.5)	2 (5)
Physical activity				
Low	25 (62.5)	22 (55)	15 (37.5)	14 (35)
Normal	12 (30)	10 (25)	20 (50)	19 (47.5)
High	3 (7.5)	8 (20)	5 (12.5)	7 (17.5)
Dietary intake				
Energy intake [kcal/d]	1423.5 ± 203.3 *	1792.4 ± 198.4 *	2290.5 ± 227.6	2439.8 ± 283.1
Proteins [g]	65.3 ± 5.1 *	76.3 ± 6.6 *	79.1 ± 6.1	83 ± 5.1
Carbohydrates [g]	200.6 ± 17.3 *	220.5 ± 20.1 *	232.4 ± 19.2	250 ± 18.2
Fats [g]	40.8 ± 3.2 *	57.3 ± 4.4 *	59.1 ± 4.7	64 ± 4.3

SD—standard deviation; TBW—total body water; FM—fat mass; FFM—fat-free mass; SMM—skeletal muscle mass; MMI—muscle mass index. * *p* < 0.05 IBD vs. healthy.

**Table 2 nutrients-17-01369-t002:** The impact of the type of disease, its duration and activity, physical activity, and dietary intake on the analyzed characteristics of the nutritional status in the study participants with IBD.

	Disease Duration [Years]		Type of Disease		Physical Activity [MET-min/Week]		Energy Intake [kcal/Day]		Proteins Intake [g/Day]		Disease Activity	
	<7	≥7	CD	UC	<600	≥600	Low	Normal	Low	Normal	Active	Remission
BMI [kg/m^2^]	23.5 ± 3.2	24.2 ± 2.9	20.9 ± 2.9 *	23.7 ± 3.1 *	24.7 ± 3.3 *	22.1 ± 3.2 *	21.2 ± 3/6 *	23.9 ± 3.4 *	23.3 ± 2.9	24.1 ± 3.3	23.2 ± 3.4	24.7 ± 4.7
Proteins [kg]	8.9 ± 2.4	8.5 ± 1.7	8.6 ± 2.1	8.8 ± 2.2	9.1 ± 1.2	8.8 ± 2.1	8.8 ± 2.1	7.8 ± 1.9	7.3 ± 1.4 *	9.9 ± 1.5	8.5 ± 1.6	8.8 ± 2.0
Minerals [kg]	5.5 ± 0.9	5.8 ± 1.4	5.2 ± 1.1	5.5 ± 1.4	5.2 ± 1.7	6.1 ± 1.1	6.4 ± 1.6	5.9 ± 1.2	5.6 ± 1.3	6.4 ± 1.8	5.4 ± 1.8	5.9 ± 1.4
TBW [kg]	39.5 ± 4.4	41.3 ± 4.9	40.9 ± 4.1	42.2 ± 4.7	40.6 ± 5.1	42.1 ± 4.4	42.1 ± 3.9	38.9 ± 4.8	39.5 ± 4.1	42.0 ± 3.4	39.1 ± 4.1	41.7 ± 4.4
FM [kg]	19.5 ± 5.1	20.6 ± 4.9	18.9 ± 4.4	20.3 ± 5.1	21.1 ± 3.5 *	18.3 ± 4.7 *	18.3 ± 3.1 *	21.9 ± 3.9 *	20.1 ± 4.1	19.9 ± 5.1	19.9 ± 4.8	20.9 ± 4.4
FM [%]	22.1 ± 3.5	22.7 ± 4.8	22.3 ± 3.2	23.2 ± 3.9	21.8 ± 3.1	22.9 ± 5.1	22.2 ± 3.7	23.1 ± 4.8	21.9 ± 3.7	22.6 ± 4.9	22.0 ± 4.4	22.9 ± 4.7
FFM [kg]	58.9 ± 4.9 *	53.2 ± 5.2 *	52.2 ± 4.1 *	58.2 ± 4.7 *	52.9 ± 4.7 *	59.1 ± 4.9 *	56.2 ± 3.9	55.7 ± 5.1	53.1 ± 3.9 *	58.9 ± 4.4 *	52.3 ± 5.5 *	58.1 ± 4.1 *
FFM [%]	75.4 ± 5.5	77.1 ± 5.8	73.7 ± 4.9 *	76.9 ± 5.1 *	71.3 ± 4.9 *	79.3 ± 5.1 *	75.5 ± 5.6	76.2 ± 4.6	75.2 ± 4.4	75.1 ± 4.9	76.1 ± 5.1	78.2 ± 5.3
SMM [kg]	38.7 ± 2.7 *	29.9 ± 3.1 *	28.9 ± 3.6 *	37.1 ± 2.9 *	30.9 ± 2.8 *	36.4 ± 3.1 *	35.9 ± 3.5	36.2 ± 2.8	30.7 ± 2.6 *	38.2 ± 2.9 *	31.5 ± 3.1 *	37.9 ± 2.6 *
MMI [kg/m^2^]	10.1 ± 2.6 *	7.9 ± 3.4 *	*7.8 ± 3.5	9.9 ± 3.1 *	8.1 ± 3.9 *	9.8 ± 3.8 *	8.4 ± 3.9	8.7 ± 3.1	8.2 ± 3.5 *	9.9 ± 3.4 *	8.1 ± 2.9 *	9.5 ± 3.6 *
Handgrip strength [kg]	27.1 ± 2.9 *	21.6 ± 3.5 *	21.9 ± 3.1 *	26.9 ± 3.3 *	22.1 ± 3.7 *	27.2 ± 2.8 *	21.8 ± 3.1 *	28.1 ± 3.2 *	22.3 ± 2.9 *	27.7 ± 3.4 *	21.6 ± 2.7 *	28.2 ± 2.9 *

BMI—body mass index; TBW—total body water; FM, fat mass; FFM—fat-free mass; SMM—skeletal muscle mass; MMI—muscle mass index. * *p* < 0.05.

**Table 3 nutrients-17-01369-t003:** Correlation between disease duration, physical activity, dietary intake, and the examined characteristics of the nutritional status of the study participants with IBD.

Characteristics	Disease Duration [Years]	Physical Activity [MET-min/Week]	Energy Intake [kcal/Day]	Proteins Intake [g/Day]
BMI [kg/m^2^]	ns	ns	0.25 *	ns
Proteins [kg]	ns	ns	ns	ns
Minerals [kg]	ns	ns	ns	ns
TBW [kg]	ns	ns	ns	ns
FM [kg]	ns	ns	ns	ns
FM [%]	ns	ns	ns	ns
FFM [kg]	−0.18 *	0.21 *	ns	0.24 *
FFM [%]	ns	0.19 *	ns	ns
SMM [kg]	−0.21 *	0.23 *	ns	0.29 *
MMI [kg/m^2^]	−0.22 *	0.22 *	ns	0.28 *
Handgrip strength [kg]	−0.24 *	0.32 *	0.24 *	0.31 *

BMI—body mass index; TBW—total body water; FM—fat mass; FFM—fat-free mass; SMM—skeletal muscle mass; MMI—muscle mass index. * *p* < 0.05; ns, not significant.

**Table 4 nutrients-17-01369-t004:** Univariate and multivariate analysis to identify factors associated with the presence of sarcopenia in IBD patients.

Characteristic	Univariate Analysis			Multivariate Analysis		
	Odds Ratio	95% CI	*p* Value	Odds Ratio	95% CI	*p* Value
Disease activity	0.514	0.334–0.874	0.0172	0.636	0.462–0.839	0.0189
Physical activity	0.421	0.308–0.945	0.0021			0.7421
Gender	0.789	0.438–0.991	0.0081	0.663	0.471–0.874	0.0191
Biologic therapy	0.481	0.412–0.997	0.0263			0.6273
Energy imbalance	0.584	0.041–1.112	0.0339			0.7229

## Data Availability

The raw data supporting the conclusions of this article will be made available by the authors on request.

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
