# Peer review of "An Assessment of the Nutritional Status in Patients with Inflammatory Bowel Disease—A Matched-Pair Case–Control Study"

_nutrients, 2025, doi:10.3390/nu17081369_

Round 1

Reviewer 1 Report

Comments and Suggestions for Authors

This matched-pair case-control study evaluates the nutritional status of patients with inflammatory bowel disease (IBD), focusing on sarcopenia, body composition, and muscle strength. The strengths include a robust matched-pair design, use of validated tools, and gender-stratified analysis. The findings highlight significant sex-specific differences, with female IBD patients exhibiting poorer nutritional status and higher sarcopenia prevalence compared to controls, while males showed reduced muscle strength without body composition differences. The study underscores the limitations of relying solely on BMI and advocates for integrating muscle strength assessments in clinical practice. However, several methodological and interpretive weaknesses limit the generalizability and depth of conclusions.

Point-by-Point Review

  1. Methodology

Sample Size and Representativeness:

The single-center design and small sample size (n=80 patients) reduce generalizability. Most participants were on biological therapy, potentially skewing results toward a subset of IBD patients with specific disease severity or treatment responses. 

Exclusion criteria (e.g., prostheses, pacemakers, limb abnormalities) may have excluded patients with comorbid conditions, limiting the representation of real-world IBD populations. The IBD information and anti-oxidative stress should be analyzed, please refer those reference(Food & Function, 2022, 13(24), 12686-12696.Food Bioscience, 50(2022), 101946.).

Control Group Limitations:

Controls were age- and sex-matched but not BMI-matched. This oversight could confound body composition comparisons, as BMI influences fat-free mass and muscle mass.

Dietary Assessment:

Reliance on self-reported 24-hour dietary recalls (three times per participant) introduces recall bias. This method is less accurate than longer-term dietary records or biomarkers.

Physical Activity Measurement:

The IPAQ questionnaire categorizes activity levels into broad MET ranges. This lacks granularity and may not capture nuanced differences in physical activity among participants.

  1. Statistical Analysis

Multiple Comparisons:

The study does not mention adjustments for multiple comparisons (e.g., Bonferroni correction). With numerous correlations tested (Table 3), the risk of Type I errors (false positives) is elevated.

Normality Testing:

While the Shapiro-Wilk test was used, the rationale for employing non-parametric tests (Mann-Whitney U) is not explicitly justified. Reporting normality outcomes would strengthen transparency.

  1. Results Interpretation

Sex-Specific Differences:

The lack of body composition differences in males is attributed to UC predominance and comparable nutrient intake. However, the discussion does not explore potential biological or hormonal factors (e.g., testosterone levels) that might protect muscle mass in males.

Disease Subtypes:

The study combines Crohn’s disease (CD) and ulcerative colitis (UC) patients but identifies significant differences between them (e.g., lower BMI in CD). Pooling these groups in analyses may obscure subtype-specific nutritional dynamics.

Causality vs. Association:

The cross-sectional design precludes causal inferences. For example, the inverse correlation between disease duration and muscle mass could reflect chronic inflammation, but confounding factors (e.g., cumulative steroid use) are not addressed.Please refer this reference(Food chemistry. 437(2024), 137834. ).

  1. Generalizability

Single-Center Bias:

Conducted at a single Polish institution, the findings may not apply to diverse populations with varying genetic, dietary, or healthcare contexts.

Treatment Heterogeneity:

Most patients were on biologics (80%), which may improve disease control and nutritional status. The impact of therapies like steroids (30.5% usage) on muscle mass is not analyzed.

  1. Discussion Limitations

Contradictory Literature:

While some cited studies align with the results (e.g., Geerling et al.), others (e.g., Prieto et al.) report conflicting findings. The discussion does not thoroughly reconcile these discrepancies, such as differences in BMI ranges or disease activity across studies.

Sarcopenia Diagnosis:

The EWGSOP2 criteria require low muscle strength and mass, yet the emphasis on handgrip strength alone (as a standalone predictor) is not critically discussed.

  1. Ethical and Reporting Issues

Incomplete Funding and Ethics Statements:

The manuscript template includes placeholder text for funding, ethics approval, and data availability. These sections must be finalized for publication readiness.

Conclusion

This study provides valuable insights into sex-specific nutritional deficits in IBD and advocates for muscle strength assessments in clinical practice. However, methodological limitations (e.g., sample size, control group matching), statistical oversights, and incomplete discussion of confounding variables necessitate cautious interpretation. Future multicenter studies with longitudinal designs, larger cohorts, and subtype-specific analyses are warranted to validate these findings.

Comments on the Quality of English Language

This matched-pair case-control study evaluates the nutritional status of patients with inflammatory bowel disease (IBD), focusing on sarcopenia, body composition, and muscle strength. The strengths include a robust matched-pair design, use of validated tools, and gender-stratified analysis. The findings highlight significant sex-specific differences, with female IBD patients exhibiting poorer nutritional status and higher sarcopenia prevalence compared to controls, while males showed reduced muscle strength without body composition differences. The study underscores the limitations of relying solely on BMI and advocates for integrating muscle strength assessments in clinical practice. However, several methodological and interpretive weaknesses limit the generalizability and depth of conclusions.

Point-by-Point Review

  1. Methodology

Sample Size and Representativeness:

The single-center design and small sample size (n=80 patients) reduce generalizability. Most participants were on biological therapy, potentially skewing results toward a subset of IBD patients with specific disease severity or treatment responses. 

Exclusion criteria (e.g., prostheses, pacemakers, limb abnormalities) may have excluded patients with comorbid conditions, limiting the representation of real-world IBD populations. The IBD information and anti-oxidative stress should be analyzed, please refer those reference(Food & Function, 2022, 13(24), 12686-12696.Food Bioscience, 50(2022), 101946.).

Control Group Limitations:

Controls were age- and sex-matched but not BMI-matched. This oversight could confound body composition comparisons, as BMI influences fat-free mass and muscle mass.

Dietary Assessment:

Reliance on self-reported 24-hour dietary recalls (three times per participant) introduces recall bias. This method is less accurate than longer-term dietary records or biomarkers.

Physical Activity Measurement:

The IPAQ questionnaire categorizes activity levels into broad MET ranges. This lacks granularity and may not capture nuanced differences in physical activity among participants.

  1. Statistical Analysis

Multiple Comparisons:

The study does not mention adjustments for multiple comparisons (e.g., Bonferroni correction). With numerous correlations tested (Table 3), the risk of Type I errors (false positives) is elevated.

Normality Testing:

While the Shapiro-Wilk test was used, the rationale for employing non-parametric tests (Mann-Whitney U) is not explicitly justified. Reporting normality outcomes would strengthen transparency.

  1. Results Interpretation

Sex-Specific Differences:

The lack of body composition differences in males is attributed to UC predominance and comparable nutrient intake. However, the discussion does not explore potential biological or hormonal factors (e.g., testosterone levels) that might protect muscle mass in males.

Disease Subtypes:

The study combines Crohn’s disease (CD) and ulcerative colitis (UC) patients but identifies significant differences between them (e.g., lower BMI in CD). Pooling these groups in analyses may obscure subtype-specific nutritional dynamics.

Causality vs. Association:

The cross-sectional design precludes causal inferences. For example, the inverse correlation between disease duration and muscle mass could reflect chronic inflammation, but confounding factors (e.g., cumulative steroid use) are not addressed.Please refer this reference(Food chemistry. 437(2024), 137834. ).

  1. Generalizability

Single-Center Bias:

Conducted at a single Polish institution, the findings may not apply to diverse populations with varying genetic, dietary, or healthcare contexts.

Treatment Heterogeneity:

Most patients were on biologics (80%), which may improve disease control and nutritional status. The impact of therapies like steroids (30.5% usage) on muscle mass is not analyzed.

  1. Discussion Limitations

Contradictory Literature:

While some cited studies align with the results (e.g., Geerling et al.), others (e.g., Prieto et al.) report conflicting findings. The discussion does not thoroughly reconcile these discrepancies, such as differences in BMI ranges or disease activity across studies.

Sarcopenia Diagnosis:

The EWGSOP2 criteria require low muscle strength and mass, yet the emphasis on handgrip strength alone (as a standalone predictor) is not critically discussed.

  1. Ethical and Reporting Issues

Incomplete Funding and Ethics Statements:

The manuscript template includes placeholder text for funding, ethics approval, and data availability. These sections must be finalized for publication readiness.

Conclusion

This study provides valuable insights into sex-specific nutritional deficits in IBD and advocates for muscle strength assessments in clinical practice. However, methodological limitations (e.g., sample size, control group matching), statistical oversights, and incomplete discussion of confounding variables necessitate cautious interpretation. Future multicenter studies with longitudinal designs, larger cohorts, and subtype-specific analyses are warranted to validate these findings.

Author Response

Dear Reviewer,
Thank you very much for your time and effort to improve the article. We have changed it according to your recommendations. 

Point-by-Point Review
1.    Methodology
Sample Size and Representativeness:

The single-center design and small sample size (n=80 patients) reduce generalizability. Most participants were on biological therapy, potentially skewing results toward a subset of IBD patients with specific disease severity or treatment responses.  
Exclusion criteria (e.g., prostheses, pacemakers, limb abnormalities) may have excluded patients with comorbid conditions, limiting the representation of real-world IBD populations. The IBD information and anti-oxidative stress should be analyzed, please refer those reference(Food & Function, 2022, 13(24), 12686-12696.Food Bioscience, 50(2022), 101946.).

We have mentioned in limitations of our study that single-center design and small sample size, as well as biological treatment in the majority of patoents reduce generalizability. The exclusion criteria (e.g., prostheses, pacemakers, limb abnormalities) were obligatory to make the BIA analysis possible. We haven’t noticed that those abnormalities were popular, in fact there are really rare. In our opinion, it hasn’t reduced generalizability.

Control Group Limitations:

Controls were age- and sex-matched but not BMI-matched. This oversight could confound body composition comparisons, as BMI influences fat-free mass and muscle mass.

It was the main aim of the study to compare the nutritional status of IBD patients to healthy people. Tha BMI-matched would limit our findings just to the body composition and wouldn’t show the fact of different BMI levels in patients and controls.

Dietary Assessment:

Reliance on self-reported 24-hour dietary recalls (three times per participant) introduces recall bias. This method is less accurate than longer-term dietary records or biomarkers.

The 24-hour dietary recalls weren’t self reported. It has been collected by the trained dietetics staff. We added the information.

Physical Activity Measurement: 
The IPAQ questionnaire categorizes activity levels into broad MET ranges. This lacks granularity and may not capture nuanced differences in physical activity among participants.

The IPAQ questionnaire is the most recommended questionnaire and those categorizes have been often used in many studies. We have tried to make the analysis in gradularity but havent’t noticed any nuanced differences. 

2.    Statistical Analysis
Multiple Comparisons:

The study does not mention adjustments for multiple comparisons (e.g., Bonferroni correction). With numerous correlations tested (Table 3), the risk of Type I errors (false positives) is elevated.

We have used adjustments for multiple comparisons – we have added the information.

3.    Results Interpretation
Sex-Specific Differences:

The lack of body composition differences in males is attributed to UC predominance and comparable nutrient intake. However, the discussion does not explore potential biological or hormonal factors (e.g., testosterone levels) that might protect muscle mass in males.

We have added the information of the potencial role of testosterone level.

4.    Generalizability
Single-Center Bias:

Conducted at a single Polish institution, the findings may not apply to diverse populations with varying genetic, dietary, or healthcare contexts.
Yes, that is the limitation of our study, which we have pointed in the limitations part of the discussion. 

Treatment Heterogeneity:

Most patients were on biologics (80%), which may improve disease control and nutritional status. The impact of therapies like steroids (30.5% usage) on muscle mass is not analyzed.

We did not find any important differences between patients using/not steroids therapies.

5.    Discussion Limitations
Contradictory Literature:

While some cited studies align with the results (e.g., Geerling et al.), others (e.g., Prieto et al.) report conflicting findings. The discussion does not thoroughly reconcile these discrepancies, such as differences in BMI ranges or disease activity across studies.
In disussion we have mentioned that the differences between studies may result from differences in study groups, as far as disease activity, diet and BMI are considered (in red). 

Sarcopenia Diagnosis: 
The EWGSOP2 criteria require low muscle strength and mass, yet the emphasis on handgrip strength alone (as a standalone predictor) is not critically discussed.

We have withdrawn this sentence. 

6.    Ethical and Reporting Issues
Incomplete Funding and Ethics Statements:
 The manuscript template includes placeholder text for funding, ethics approval, and data availability. These sections must be finalized for publication readiness.

We have added the information.

Conclusion
This study provides valuable insights into sex-specific nutritional deficits in IBD and advocates for muscle strength assessments in clinical practice. However, methodological limitations (e.g., sample size, control group matching), statistical oversights, and incomplete discussion of confounding variables necessitate cautious interpretation. Future multicenter studies with longitudinal designs, larger cohorts, and subtype-specific analyses are warranted to validate these findings.

We have emphasised this conclusion.

Reviewer 2 Report

Comments and Suggestions for Authors

Assessment of the Nutritional Status in Patients with Inflammatory Bowel Disease. A Matched-Pair Case-Control Study

   This study is interesting because using a simplified and objective methods (electrical bioimpedance methods) early changes of body composition (sarcopenia) or nutritional status (muscle mass, muscle strength, muscle mass index) and hand grip strength have been evaluated meticulously. Author has elucidated several components of nutritional elements such as BMI, muscle mass, and nutritional status.

Author stated as follows: “In our study, the patients with IBD were characterized by a poorer nutritional status than the healthy subjects, mainly in terms of fat-free body mass and muscle mass, and consequently a higher incidence of sarcopenia, especially in the female group. In the diagnosis of sarcopenia, clinical assessment based on BMI alone is insufficient, therefore muscle strength testing should be included in the routine anthropometric evaluation of patients with IBD because of its association with low muscle mass.”

Results of these metrics several components have been introduced but which is key index representing nutritional status and which index is most representative sarcopenia? Author should determine these questions. Thereafter, there are several confounding factors which affect causatively. Therefore, author should perform multiple regression analysis to elucidate the key factor which affect nutritional status (sex, duration of illness, remission or active, nutritional intake imbalance, steroid intake or so on).

Author Response

This study is interesting because using a simplified and objective methods (electrical bioimpedance methods) early changes of body composition (sarcopenia) or nutritional status (muscle mass, muscle strength, muscle mass index) and hand grip strength have been evaluated meticulously. Author has elucidated several components of nutritional elements such as BMI, muscle mass, and nutritional status.
Author stated as follows: “In our study, the patients with IBD were characterized by a poorer nutritional status than the healthy subjects, mainly in terms of fat-free body mass and muscle mass, and consequently a higher incidence of sarcopenia, especially in the female group. In the diagnosis of sarcopenia, clinical assessment based on BMI alone is insufficient, therefore muscle strength testing should be included in the routine anthropometric evaluation of patients with IBD because of its association with low muscle mass.”
Results of these metrics several components have been introduced but which is key index representing nutritional status and which index is most representative sarcopenia? Author should determine these questions. Thereafter, there are several confounding factors which affect causatively. Therefore, author should perform multiple regression analysis to elucidate the key factor which affect nutritional status (sex, duration of illness, remission or active, nutritional intake imbalance, steroid intake or so on).

Dear Reviewer,
Thank you very much for your time and effort to improve the article. We have changed it according to your recommendations. 
We have added the multivariate analyses to identify factors associated with sarcopenia.

Reviewer 3 Report

Comments and Suggestions for Authors

I read this manuscript by Godala et al. with interest.

I have a few considerations about it:

  1. In my opinion, the abstract is rather lengthy and should be streamlined and preferably presented in a structured format to enhance data readability.
  2. I believe that the introduction should specify that patients with IBD often engage in reduced regular physical activity due to fear of disease exacerbation and, frequently, a lack of encouragement from their social network. This serves as an additional barrier to improving sarcopenia. I recommend supporting this statement by citing: https://pubmed.ncbi.nlm.nih.gov/38723769/. Moreover, this study also utilised the IPAQ, as you did.
  3. I would also suggest adding in the introduction that new findings are emerging in the pathogenesis of IBD concerning the melanocortin system. This system acts as a regulator of lean and fat mass and is often used as a therapeutic approach for metabolic disorders related to body weight. I would cite, for instance, the emerging role of MC3R and MC5R receptors as indicators of histological disease activity.
  4. Using only an endoscopic diagnosis of IBD in the methods section is risky because many conditions can completely mimic the endoscopic appearance of IBD. Are you certain it would not be more appropriate to specify, if applicable, a histological diagnosis as well?
  5. The exclusion criteria, in my opinion, should be better explained, including the rationale for excluding those specific patients.
  6. Patient enrolment numbers should not be mentioned in the methods section, as this belongs to the results (refer to STROBE checklists).
  7. The statistical analysis section appears rather succinct and should be elaborated upon. Additionally, considering continuous variables as normally distributed (allowing for the use of means and SD instead of medians and IQR) seems somewhat risky given a total sample size of only 160 patients.
  8. I encourage the authors to consider including some figures to present the most significant results; the exclusive use of tables and text makes the data quite cumbersome to read.

Author Response

Dear Reviewer,

Thank you very much for your time and effort. We have tried to change the article according to your suggestions.

•  In my opinion, the abstract is rather lengthy and should be streamlined and preferably presented in a structured format to enhance data readability. 

We have made the abstract shorter.

•  I believe that the introduction should specify that patients with IBD often engage in reduced regular physical activity due to fear of disease exacerbation and, frequently, a lack of encouragement from their social network. This serves as an additional barrier to improving sarcopenia. I recommend supporting this statement by citing: https://pubmed.ncbi.nlm.nih.gov/38723769/. Moreover, this study also utilised the IPAQ, as you did. 

We have improved the discussion and cited suggested article.

•  I would also suggest adding in the introduction that new findings are emerging in the pathogenesis of IBD concerning the melanocortin system. This system acts as a regulator of lean and fat mass and is often used as a therapeutic approach for metabolic disorders related to body weight. I would cite, for instance, the emerging role of MC3R and MC5R receptors as indicators of histological disease activity. 

We have added suggested information. [Gravina AG, Pellegrino R, Durante T, Palladino G, Imperio G, D'Amico G, Trotta MC, Dallio M, Romeo M, D'Amico M, Federico A. The Melanocortin System in Inflammatory Bowel Diseases: Insights into Its Mechanisms and Therapeutic Potentials. Cells. 2023 Jul 19;12(14):1889. doi: 10.3390/cells12141889.]

•  Using only an endoscopic diagnosis of IBD in the methods section is risky because many conditions can completely mimic the endoscopic appearance of IBD. Are you certain it would not be more appropriate to specify, if applicable, a histological diagnosis as well? 

Histological diagnosis has been done. We added information in methods section 

•  The exclusion criteria, in my opinion, should be better explained, including the rationale for excluding those specific patients. 

We have added more information.

•  Patient enrolment numbers should not be mentioned in the methods section, as this belongs to the results (refer to STROBE checklists).

We have withdrawn the numbers.

•  The statistical analysis section appears rather succinct and should be elaborated upon. Additionally, considering continuous variables as normally distributed (allowing for the use of means and SD instead of medians and IQR) seems somewhat risky given a total sample size of only 160 patients. 

With all due respect, statistical analysis was made by the professional staff. We do not feel that it appears succinct.

Round 2

Reviewer 1 Report

Comments and Suggestions for Authors

The manuscript presents a matched-pair case-control study assessing nutritional status, body composition, and sarcopenia in patients with inflammatory bowel disease (IBD). The study highlights significant gender-specific differences, with female IBD patients exhibiting lower muscle mass, BMI, and higher sarcopenia prevalence compared to healthy controls, while male patients showed no significant differences except reduced muscle strength. The use of bioelectrical impedance analysis (BIA) and handgrip strength measurements aligns with current sarcopenia diagnostic criteria (EWGSOP2), and the inclusion of dietary intake and physical activity data adds depth. However, limitations such as a single-center design, small sample size for subgroup analyses, and reliance on BIA instead of gold-standard imaging methods temper the generalizability and precision of findings. The discussion contextualizes results within existing literature but overlooks potential confounding factors like medication effects. Despite these issues, the study underscores the importance of integrating muscle strength assessments into routine IBD care, particularly for women.

Point-by-Point Review

  1. Study Design and Sample

Single-center design limits generalizability to broader populations. Small sample size (n=80 IBD patients) reduces power for subgroup analyses (e.g., CD vs. UC, active vs. remission).

Control group selection: Healthy controls may not account for lifestyle/behavioral confounders (e.g., stress, comorbidities). Please refer and cite those references(Food Chemistry,472 (2025),142932.Food Bioscience, 50(2022), 101946. ).

  1. Methodology

BIA limitations: While practical, BIA is less accurate than DXA/CT for body composition, potentially misclassifying sarcopenia.

Self-reported physical activity: IPAQ is prone to recall bias.No inflammation biomarkers: CRP or fecal calprotectin levels were not measured to correlate with muscle loss.

  1. Statistical Analysis

Incomplete adjustment: Key confounders (e.g., corticosteroid use, disease location) were not controlled in multivariate models.

Correlation vs. causation: Cross-sectional design precludes causal inferences (e.g., disease duration vs. muscle loss).

  1. Results

Lack of male subgroup insights: No exploration of why men showed no differences in body composition despite lower muscle strength.

Inconsistent definitions: MMI cutoff values for sarcopenia (e.g., women <6.2 kg/m²) may not align with other studies, complicating comparisons.

  1. Discussion

Overlooked confounders: Medications (e.g., biologics, steroids) and hormonal differences (e.g., testosterone) were mentioned but not analyzed.

Redundancy: Repetitive comparisons with literature (e.g., Geerling, Valentini).

Inadequate mechanistic exploration: No discussion of inflammation’s role in muscle catabolism.

  1. Tables and Presentation

Table readability: Overly dense tables (e.g., Table 2) could be simplified or split for clarity.

Missing data: No stratification of results by IBD type (CD vs. UC) in key analyses.

  1. Ethical and Practical Considerations

Lack of open data: No statement on data availability for replication.

Key Weaknesses Summary

Limited generalizability due to single-center design and small sample. Reliance on BIA rather than gold-standard imaging for body composition.

Incomplete adjustment for confounders (medications, inflammation).

Cross-sectional design limits causal conclusions.

Underpowered subgroup analyses (e.g., CD vs. UC, males vs. females).

Recommendations for Revision

Address confounders (e.g., medications, hormones) in multivariate models.

Discuss limitations of BIA more critically.

Simplify tables and stratify results by IBD type.

Propose future multi-center studies with longitudinal design and biomarker integration.

Comments on the Quality of English Language

The manuscript presents a matched-pair case-control study assessing nutritional status, body composition, and sarcopenia in patients with inflammatory bowel disease (IBD). The study highlights significant gender-specific differences, with female IBD patients exhibiting lower muscle mass, BMI, and higher sarcopenia prevalence compared to healthy controls, while male patients showed no significant differences except reduced muscle strength. The use of bioelectrical impedance analysis (BIA) and handgrip strength measurements aligns with current sarcopenia diagnostic criteria (EWGSOP2), and the inclusion of dietary intake and physical activity data adds depth. However, limitations such as a single-center design, small sample size for subgroup analyses, and reliance on BIA instead of gold-standard imaging methods temper the generalizability and precision of findings. The discussion contextualizes results within existing literature but overlooks potential confounding factors like medication effects. Despite these issues, the study underscores the importance of integrating muscle strength assessments into routine IBD care, particularly for women.

Point-by-Point Review

  1. Study Design and Sample

Single-center design limits generalizability to broader populations. Small sample size (n=80 IBD patients) reduces power for subgroup analyses (e.g., CD vs. UC, active vs. remission).

Control group selection: Healthy controls may not account for lifestyle/behavioral confounders (e.g., stress, comorbidities). Please refer and cite those references(Food Chemistry,472 (2025),142932.Food Bioscience, 50(2022), 101946. ).

  1. Methodology

BIA limitations: While practical, BIA is less accurate than DXA/CT for body composition, potentially misclassifying sarcopenia.

Self-reported physical activity: IPAQ is prone to recall bias.No inflammation biomarkers: CRP or fecal calprotectin levels were not measured to correlate with muscle loss.

  1. Statistical Analysis

Incomplete adjustment: Key confounders (e.g., corticosteroid use, disease location) were not controlled in multivariate models.

Correlation vs. causation: Cross-sectional design precludes causal inferences (e.g., disease duration vs. muscle loss).

  1. Results

Lack of male subgroup insights: No exploration of why men showed no differences in body composition despite lower muscle strength.

Inconsistent definitions: MMI cutoff values for sarcopenia (e.g., women <6.2 kg/m²) may not align with other studies, complicating comparisons.

  1. Discussion

Overlooked confounders: Medications (e.g., biologics, steroids) and hormonal differences (e.g., testosterone) were mentioned but not analyzed.

Redundancy: Repetitive comparisons with literature (e.g., Geerling, Valentini).

Inadequate mechanistic exploration: No discussion of inflammation’s role in muscle catabolism.

  1. Tables and Presentation

Table readability: Overly dense tables (e.g., Table 2) could be simplified or split for clarity.

Missing data: No stratification of results by IBD type (CD vs. UC) in key analyses.

  1. Ethical and Practical Considerations

Lack of open data: No statement on data availability for replication.

Key Weaknesses Summary

Limited generalizability due to single-center design and small sample. Reliance on BIA rather than gold-standard imaging for body composition.

Incomplete adjustment for confounders (medications, inflammation).

Cross-sectional design limits causal conclusions.

Underpowered subgroup analyses (e.g., CD vs. UC, males vs. females).

Recommendations for Revision

Address confounders (e.g., medications, hormones) in multivariate models.

Discuss limitations of BIA more critically.

Simplify tables and stratify results by IBD type.

Propose future multi-center studies with longitudinal design and biomarker integration.

Author Response

Dear Reviewer, thank you very much for your time and effort. We have tried to take into account all your recommendations, but some od them are impossible to do, because we can not change the study, which is already done. Despite of your suggestions that can not be done, we believe that the results of our study are worth of publication. 

Point-by-Point Review
1.    Study Design and Sample
Single-center design limits generalizability to broader populations. Small sample size (n=80 IBD patients) reduces power for subgroup analyses (e.g., CD vs. UC, active vs. remission).
Control group selection: Healthy controls may not account for lifestyle/behavioral confounders (e.g., stress, comorbidities). Please refer and cite those references(Food Chemistry,472 (2025),142932.Food Bioscience, 50(2022), 101946. ).
We have mentioned in limitations of the study that control group was based on healthy people. We get the feeling that the Reviewer try to force us to cite references, which is forbidden by the redaction.

2.    Methodology
BIA limitations: While practical, BIA is less accurate than DXA/CT for body composition, potentially misclassifying sarcopenia.
Self-reported physical activity: IPAQ is prone to recall bias.No inflammation biomarkers: CRP or fecal calprotectin levels were not measured to correlate with muscle loss.
We have discussed the limitations of BIA method, but also highlighted its importance in daily clinical practice (DXA is inconvienient and expensive). IPAQ is strongly recommended method, very often used in studies. The correlation between CRP with muscle mass did not bring any important results. 

3.    Statistical Analysis
Incomplete adjustment: Key confounders (e.g., corticosteroid use, disease location) were not controlled in multivariate models.
Correlation vs. causation: Cross-sectional design precludes causal inferences (e.g., disease duration vs. muscle loss).
We did not find any correlations between suggested markers.

4.    Results
Lack of male subgroup insights: No exploration of why men showed no differences in body composition despite lower muscle strength.
Inconsistent definitions: MMI cutoff values for sarcopenia (e.g., women <6.2 kg/m²) may not align with other studies, complicating comparisons.
In discussion we have tried to explain why men showed no differences in body composition despite lower muscle strength. We used MMI, because it is recommended indicator for BIA method of body composition. 

5.    Discussion
Overlooked confounders: Medications (e.g., biologics, steroids) and hormonal differences (e.g., testosterone) were mentioned but not analyzed.
Redundancy: Repetitive comparisons with literature (e.g., Geerling, Valentini).
Inadequate mechanistic exploration: No discussion of inflammation’s role in muscle catabolism. 

6.    Tables and Presentation
Table readability: Overly dense tables (e.g., Table 2) could be simplified or split for clarity.
Missing data: No stratification of results by IBD type (CD vs. UC) in key analyses.
The stratification of result by IBD type 

7.    Ethical and Practical Considerations
Lack of open data: No statement on data availability for replication.
We added the statement.

Reviewer 2 Report

Comments and Suggestions for Authors

In the revised manuscript multivariate analysis was conducted but the results were not intoduced in the conclusions and in abstract. This multivariate analysis has shown that patients with active disease had a relatively strong sarcopenia and women had tendency to be impaired nutritional status or sracopenia. This will suggest only that patient selection was not appropriate. These comments should be properly incorprated into abstract and conclusions.

Author Response

Dear Reviewer,
Thank you for your time. We have added the missing information into the abstract and conclusions.

Reviewer 3 Report

Comments and Suggestions for Authors

The authors have replied to all comments. 

Round 3

Reviewer 1 Report

Comments and Suggestions for Authors

The manuscript presents a matched-pair case-control study assessing nutritional status and sarcopenia in patients with inflammatory bowel disease (IBD) using bioelectrical impedance analysis (BIA) and handgrip strength. The study highlights significant gender-specific differences, with female IBD patients exhibiting lower muscle mass, BMI, and higher sarcopenia prevalence compared to healthy controls, while male patients showed differences only in muscle strength. The work addresses a clinically relevant topic, employs validated tools, and integrates multiple factors (disease activity, dietary intake, physical activity).

However, limitations in sample size, generalizability, and methodological design reduce the robustness of conclusions. The findings underscore the need for comprehensive nutritional assessments in IBD but require cautious interpretation due to identified weaknesses.

Study Design and Sample Size

Small sample size (80 patients/controls) limits power, particularly for subgroup analyses (e.g., men vs. women, CD vs. UC). Single-center recruitment (Poland) restricts generalizability to diverse populations.

Control Group Selection

Controls were not matched for BMI, a critical factor in nutritional studies. This omission may confound comparisons, as BMI influences body composition metrics.

Methodological Considerations

Physical activity data relied on self-reported IPAQ, introducing recall bias. No objective measures (e.g., accelerometry) were used to validate activity levels.

Medications (e.g., biologics, steroids) were not adjusted for in multivariate analyses, despite their potential impact on muscle mass and inflammation. Please refer this reference(Food Chemistry,472 (2025),142932.).

Biological mechanisms (e.g., hormonal influences) for these disparities were not explored, leaving findings descriptive rather than explanatory.

Disease Heterogeneity

CD and UC were combined in analyses despite pathophysiological differences. Subgroup analyses for disease type were limited, reducing clinical applicability.

Dietary Assessment

Dietary intake was assessed via 24-hour recalls, which may not reflect habitual intake. No validation against biomarkers (e.g., nitrogen balance) was performed.

Discussion of Contradictory Literature

Superficial exploration of reasons for contradictions (e.g., regional dietary habits, treatment protocols). Please refer those reference(Food Bioscience, 50(2022), 101946.Food & Function, 2022, 13(24), 12686-12696.).

Clinical Implications

Fails to propose specific interventions (e.g., protein supplementation, exercise programs) to address sarcopenia.

Supplementary Materials and Data Presentation

Tables referenced in the text were not included in the provided content, hindering full evaluation of data quality and presentation.

Recommendations for Improvement:

Increase sample size and multicenter collaboration to enhance generalizability.

Match controls for BMI or adjust analyses for BMI disparities.

Incorporate objective physical activity measurements and adjust for medication effects.

Explore biological mechanisms behind gender differences.

Provide subgroup-specific recommendations for CD vs. UC patients.

Include missing tables/supplementary materials for transparency.

This study provides valuable insights into IBD-related sarcopenia but requires methodological refinements and deeper mechanistic exploration to strengthen its impact.

Comments on the Quality of English Language

The manuscript presents a matched-pair case-control study assessing nutritional status and sarcopenia in patients with inflammatory bowel disease (IBD) using bioelectrical impedance analysis (BIA) and handgrip strength. The study highlights significant gender-specific differences, with female IBD patients exhibiting lower muscle mass, BMI, and higher sarcopenia prevalence compared to healthy controls, while male patients showed differences only in muscle strength. The work addresses a clinically relevant topic, employs validated tools, and integrates multiple factors (disease activity, dietary intake, physical activity).

However, limitations in sample size, generalizability, and methodological design reduce the robustness of conclusions. The findings underscore the need for comprehensive nutritional assessments in IBD but require cautious interpretation due to identified weaknesses.

Study Design and Sample Size

Small sample size (80 patients/controls) limits power, particularly for subgroup analyses (e.g., men vs. women, CD vs. UC). Single-center recruitment (Poland) restricts generalizability to diverse populations.

Control Group Selection

Controls were not matched for BMI, a critical factor in nutritional studies. This omission may confound comparisons, as BMI influences body composition metrics.

Methodological Considerations

Physical activity data relied on self-reported IPAQ, introducing recall bias. No objective measures (e.g., accelerometry) were used to validate activity levels.

Medications (e.g., biologics, steroids) were not adjusted for in multivariate analyses, despite their potential impact on muscle mass and inflammation. Please refer this reference(Food Chemistry,472 (2025),142932.).

Biological mechanisms (e.g., hormonal influences) for these disparities were not explored, leaving findings descriptive rather than explanatory.

Disease Heterogeneity

CD and UC were combined in analyses despite pathophysiological differences. Subgroup analyses for disease type were limited, reducing clinical applicability.

Dietary Assessment

Dietary intake was assessed via 24-hour recalls, which may not reflect habitual intake. No validation against biomarkers (e.g., nitrogen balance) was performed.

Discussion of Contradictory Literature

Superficial exploration of reasons for contradictions (e.g., regional dietary habits, treatment protocols). Please refer those reference(Food Bioscience, 50(2022), 101946.Food & Function, 2022, 13(24), 12686-12696.).

Clinical Implications

Fails to propose specific interventions (e.g., protein supplementation, exercise programs) to address sarcopenia.

Supplementary Materials and Data Presentation

Tables referenced in the text were not included in the provided content, hindering full evaluation of data quality and presentation.

Recommendations for Improvement:

Increase sample size and multicenter collaboration to enhance generalizability.

Match controls for BMI or adjust analyses for BMI disparities.

Incorporate objective physical activity measurements and adjust for medication effects.

Explore biological mechanisms behind gender differences.

Provide subgroup-specific recommendations for CD vs. UC patients.

Include missing tables/supplementary materials for transparency.

This study provides valuable insights into IBD-related sarcopenia but requires methodological refinements and deeper mechanistic exploration to strengthen its impact.

Author Response

Dear Reviewer,
Thank you very much for your time and effort to improve the article. We have changed it according to your recommendations. 

Point-by-Point Review
1.    Methodology
Sample Size and Representativeness:

The single-center design and small sample size (n=80 patients) reduce generalizability. Most participants were on biological therapy, potentially skewing results toward a subset of IBD patients with specific disease severity or treatment responses.  
Exclusion criteria (e.g., prostheses, pacemakers, limb abnormalities) may have excluded patients with comorbid conditions, limiting the representation of real-world IBD populations. The IBD information and anti-oxidative stress should be analyzed, please refer those reference(Food & Function, 2022, 13(24), 12686-12696.Food Bioscience, 50(2022), 101946.).
 We have mentioned in limitations of our study that single-center design and small sample size, as well as biological treatment in the majority of patoents reduce generalizability. The exclusion criteria (e.g., prostheses, pacemakers, limb abnormalities) were obligatory to make the BIA analysis possible. We haven’t noticed that those abnormalities were popular, in fact there are really rare. In our opinion, it hasn’t reduced generalizability.

Control Group Limitations:

Controls were age- and sex-matched but not BMI-matched. This oversight could confound body composition comparisons, as BMI influences fat-free mass and muscle mass.
It was the main aim of the study to compare the nutritional status of IBD patients to healthy people. Tha BMI-matched would limit our findings just to the body composition and wouldn’t show the fact of different BMI levels in patients and controls.

Dietary Assessment:

Reliance on self-reported 24-hour dietary recalls (three times per participant) introduces recall bias. This method is less accurate than longer-term dietary records or biomarkers.
The 24-hour dietary recalls weren’t self reported. It has been collected by the trained dietetics staff. We added the information.

Physical Activity Measurement: 
The IPAQ questionnaire categorizes activity levels into broad MET ranges. This lacks granularity and may not capture nuanced differences in physical activity among participants.
The IPAQ questionnaire is the most recommended questionnaire and those categorizes have been often used in many studies. We have tried to make the analysis in gradularity but havent’t noticed any nuanced differences. 

2.    Statistical Analysis
Multiple Comparisons:

The study does not mention adjustments for multiple comparisons (e.g., Bonferroni correction). With numerous correlations tested (Table 3), the risk of Type I errors (false positives) is elevated.
We have used adjustments for multiple comparisons – we have added the information.

3.    Results Interpretation
Sex-Specific Differences:

The lack of body composition differences in males is attributed to UC predominance and comparable nutrient intake. However, the discussion does not explore potential biological or hormonal factors (e.g., testosterone levels) that might protect muscle mass in males.
We have added the information of the potencial role of testosterone level.
Causality vs. Association:

The cross-sectional design precludes causal inferences. For example, the inverse correlation between disease duration and muscle mass could reflect chronic inflammation, but confounding factors (e.g., cumulative steroid use) are not addressed.Please refer this reference(Food chemistry. 437(2024), 137834. ).

4.    Generalizability
Single-Center Bias:

Conducted at a single Polish institution, the findings may not apply to diverse populations with varying genetic, dietary, or healthcare contexts.
Yes, that is the limitation of our study, which we have pointed in the limitations part of the discussion. 

Treatment Heterogeneity:

Most patients were on biologics (80%), which may improve disease control and nutritional status. The impact of therapies like steroids (30.5% usage) on muscle mass is not analyzed.
We did not find any important differences between patients using/not steroids therapies.

5.    Discussion Limitations
Contradictory Literature:

While some cited studies align with the results (e.g., Geerling et al.), others (e.g., Prieto et al.) report conflicting findings. The discussion does not thoroughly reconcile these discrepancies, such as differences in BMI ranges or disease activity across studies.
In disussion we have mentioned that the differences between studies may result from differences in study groups, as far as disease activity, diet and BMI are considered (in red). 

Sarcopenia Diagnosis: 
The EWGSOP2 criteria require low muscle strength and mass, yet the emphasis on handgrip strength alone (as a standalone predictor) is not critically discussed.
We have withdrawn this sentence. 
6.    Ethical and Reporting Issues
Incomplete Funding and Ethics Statements:
 The manuscript template includes placeholder text for funding, ethics approval, and data availability. These sections must be finalized for publication readiness.
We have added the information.

Conclusion
This study provides valuable insights into sex-specific nutritional deficits in IBD and advocates for muscle strength assessments in clinical practice. However, methodological limitations (e.g., sample size, control group matching), statistical oversights, and incomplete discussion of confounding variables necessitate cautious interpretation. Future multicenter studies with longitudinal designs, larger cohorts, and subtype-specific analyses are warranted to validate these findings.
We have emphasised this conclusion.

With all due respect, the article has been translated by the professional staff. We can not agree that the quality of English is poor.

Reviewer 2 Report

Comments and Suggestions for Authors

Now this re-revised manuscript has been amended comprehensively fully. 

Author Response

Thank you!